# Genome sequences of two diploid wild relatives of cultivated sweetpotato reveal targets for genetic improvement

Shan Wu [1], Kin H. Lau [2], Qinghe Cao[1,3], John P. Hamilton [2], Honghe Sun[1,4], Chenxi Zhou [5], Lauren Eserman[6,17], Dorcus C. Gemenet [7], Bode A. Olukolu[8,9], Haiyan Wang[2,10], Emily Crisovan[2], Grant T. Godden[2], Chen Jiao[1], Xin Wang[1], Mercy Kitavi [11], Norma Manrique-Carpintero[2], Brieanne Vaillancourt[2], Krystle Wiegert-Rininger [2], Xinsun Yang[12], Kan Bao[1], Jennifer Schaff[13], Jan Kreuze [7], Wolfgang Gruneberg[7], Awais Khan[7,18], Marc Ghislain[11], Daifu Ma[3], Jiming Jiang[2,10], Robert O.M. Mwanga[14], Jim Leebens-Mack [6], Lachlan J.M. Coin [5], G. Craig Yencho[8], C. Robin Buell[2,15] & Zhangjun Fei [1,16]

Sweetpotato [*Ipomoea batatas* (L.) Lam.] is a globally important staple food crop, especially for sub-Saharan Africa. Agronomic improvement of sweetpotato has lagged behind other major food crops due to a lack of genomic and genetic resources and inherent challenges in breeding a heterozygous, clonally propagated polyploid. Here, we report the genome sequences of its two diploid relatives, *I. trifida* and *I. triloba*, and show that these high-quality genome assemblies are robust references for hexaploid sweetpotato. Comparative and phylogenetic analyses reveal insights into the ancient whole-genome triplication history of *Ipomoea* and evolutionary relationships within the Batatas complex. Using resequencing data from 16 genotypes widely used in African breeding programs, genes and alleles associated with carotenoid biosynthesis in storage roots are identified, which may enable efficient breeding of varieties with high provitamin A content. These resources will facilitate genome-enabled breeding in this important food security crop.

[1] Boyce Thompson Institute, Cornell University, Ithaca, NY 14853, USA. [2] Department of Plant Biology, Michigan State University, East Lansing, MI 48824, USA. [3] Jiangsu Xuzhou Sweetpotato Research Center, Xuzhou, Jiangsu 221131, China. [4] National Engineering Research Center for Vegetables, Beijing Academy of Agriculture and Forestry Sciences, Beijing 100097, China. [5] Institute for Molecular Bioscience, University of Queensland, St Lucia, Brisbane, QLD 4072, Australia. [6] Department of Plant Biology, University of Georgia, Athens, GA 30602, USA. [7] International Potato Center, Lima 12, Peru. [8] Department of Horticultural Science, North Carolina State University, Raleigh, NC 27695, USA. [9] Department of Entomology and Plant Pathology, University of Tennessee, Knoxville, TN 37996, USA. [10] Department of Horticulture, Michigan State University, East Lansing, MI 48824, USA. [11] International Potato Center, Nairobi 00603, Kenya. [12] Food Crops Institute, Hubei Academy of Agricultural Sciences, Wuhan 430064, China. [13] Genomic Sciences Laboratory, North Carolina State University, Raleigh, NC 27695, USA. [14] International Potato Center, Kampala, Uganda. [15] Plant Resilience Institute, Michigan State University, East Lansing, MI 48824, USA. [16] USDA-ARS Robert W. Holley Center for Agriculture and Health, Ithaca, NY 14853, USA. [17] Present address: Department of Conservation and Research, Atlanta Botanical Garden, Atlanta, GA 30309, USA. [18] Present address: Plant Pathology and Plant-Microbe Biology Section, Cornell University, Geneva, NY 14456, USA. These authors contributed equally: Shan Wu, Kin H. Lau, Qinghe Cao, John P. Hamilton, Honghe Sun. Correspondence and requests for materials should be addressed to C.R.B. (email: buell@msu.edu) or to Z.F. (email: zf25@cornell.edu)

Sweetpotato, *Ipomoea batatas* (L.) Lam. ($2n = 6x = 90$), is an important food crop ranking seventh globally with 106.6 million tons produced in 2014 (FAOSTAT 2014). Sweetpotato provides a rich source of carbohydrates, dietary fiber, vitamins, and micronutrients, is low in fat and cholesterol[1], and due to its resilience and adaptability, it serves an important role in food security for subsistence farmers in sub-Saharan Africa (SSA). Indeed, efforts by the International Potato Center (CIP) to replace the dominant β-carotene lacking white-fleshed varieties in SSA with provitamin A-rich orange-fleshed sweetpotato (OFSP)[2] were considered a breakthrough achievement as vitamin A deficiency affects more than 40% of children under 5 years old in SSA and is a leading cause of blindness and premature death[3]. Awarding of the World Food Prize in 2016 to four scientists who pioneered biofortification with OFSP highlights the significance of these efforts and the importance of OFSP in shifting human health outcomes.

Sweetpotato was domesticated in tropical America at least 5000 years ago in a region between the Yucatan Peninsula of Mexico and the Orinoco River in Venezuela[4], and introduced to Europe and Africa from South America by Europeans in the early 16th century and later to the rest of the world[5]. Whereas sweetpotato has long been thought to have been introduced to Oceania by Polynesians in pre-Columbian times[5], this hypothesis has recently been challenged[6]. Sweetpotato is one of only two food crops (the other being water spinach, *I. aquatic* Forssk.) among ~1600 species in the morning glory family (Convolvulaceae). The wild diploid species *I. trifida* (Kunth.) G. Don ($2n = 2x = 30$) has been widely reported as the closest relative of cultivated sweetpotato[6,7], but there are two hypotheses regarding the origin of the cultivated hexaploid sweetpotato: an autopolyploidization within an *I. trifida* progenitor population[6,7] or an allo-autopolyploidization involving both *I. trifida* and another diploid wild relative, possibly *Ipomoea triloba* L. ($2n = 2x = 30$)[4]. Some accessions of *I. trifida* are sexually compatible with *I. batatas*, though this is rare and, unlike *I. triloba*, commonly produces unreduced pollen associated with polyploidization[8,9]. Observations of a predominance of polysomic inheritance have been interpreted as support for the autopolyploidization hypothesis[10]. However, frequent tetrasomic recombination has been observed in allopolyploids, such as cultivated peanut (*Arachis hypogaea*)[11]. Some molecular data implicate *I. trifida* as a close, but not direct ancestor, and provide no evidence for *I. triloba* as contributing to the hexaploid genome of sweetpotato[7,12]. Roullier et al.[7] speculate that sweetpotato could share a common ancestor with *I. trifida* and/or tetraploid *I. batatas* yet have multiple origins with at least two autopolyploidization events from a single progenitor species. Munoz-Rodrıguez et al.[6] recently used targeted gene capture to analyze hexaploid *I. batatas* and wild relatives in an attempt to resolve the origin of the hexaploid sweetpotato, but their findings have yet to be tested using whole-genome sequence data.

The improvement of sweetpotato varieties across the world faces major constraints due to the lack of knowledge of the genetic, molecular, and physiological basis of key agronomic traits of this critical food crop. The sweetpotato genome is large (1.6 Gb)[13] and complex due to polyploidy and a high degree of heterozygosity, challenging efforts to generate a high-quality de novo genome assembly. While an assembly of *I. trifida* was released in 2015, it was highly fragmented[14], and thus its use as a reference for genome-enabled breeding efforts in sweetpotato is limited. Likewise, a recently reported haplotype-resolved genome assembly of *I. batatas*[15] is incomplete and contains a considerable amount of redundancy and misassemblies, including erroneous assembly of haplotypes. To provide high-quality reference genome resources and further investigate the relationship between hexaploid sweetpotato and its diploid relatives, we generated high quality, chromosome-scale genome assemblies of *I. trifida* and *I. triloba*, resequenced 16 cultivars and landraces widely used in African breeding programs, and performed a suite of comparative genomic analyses. Our findings illustrate the utility of high-quality diploid *I. trifida* and *I. triloba* genomes for genome-enabled breeding and research in this important food security crop.

## Results

**Genome assemblies of *I. trifida* and *I. triloba*.** We sequenced the genomes of *I. trifida* NCNSP0306 and *I. triloba* NCNSP0323, two diploid species closely related to the hexaploid sweetpotato (Fig. 1 and Supplementary Method 1). A total of 104.8 and 144.1 Gb high-quality cleaned Illumina paired-end and mate-pair reads were generated for *I. trifida* and *I. triloba*, respectively, representing 199× and 291× coverage of their genomes (Supplementary Table 1). Based on the 17-mer depth distribution of the Illumina reads (Supplementary Fig. 1), the *I. trifida* genome is 526.4 Mb with a heterozygosity level of 0.24% calculated from single nucleotide polymorphism (SNP) analyses. In contrast, *I. triloba* NCNSP0323 is highly homozygous (Supplementary Fig. 1) with an estimated genome size of 495.9 Mb. Long-read PacBio sequence data (11× and 5× for *I. trifida* and *I. triloba*, respectively; Supplementary Table 1) was used for gap-filling and used in combination with de novo-assembled BioNano genome maps to improve the assembly (Supplementary Method 2). The resulting high-quality assemblies for *I. trifida* and *I. triloba* were 462.0 and 457.8 Mb with scaffold N50 lengths of 1.2 and 6.9 Mb, respectively (Supplementary Table 2).

To construct pseudomolecules, a high-density genetic map for *I. trifida* was generated using an F₁ population derived from a cross between two heterozygous *I. trifida* lines, M9 and M19 (Supplementary Method 3 and Supplementary Fig. 2). In all, 461 *I. trifida* scaffolds with a cumulative size of 373.4 Mb (80.8% of the assembly) were anchored to 15 linkage groups (LGs), of which 196 (66.3%) were oriented (Supplementary Table 3). The *I. trifida* genetic map was also used to anchor 216 *I. triloba* scaffolds (443.3 Mb; 96.8% of the assembly) to the 15 LGs (Supplementary Table 3).

To evaluate the quality of the assemblies, RNA-Seq reads from a set of developmental tissues were aligned to their cognate assemblies; 85.3% and 91.4% of the reads aligned to the *I. trifida* and *I. triloba* genomes, respectively (Supplementary Data 1). The completeness of the assemblies in terms of gene content was assessed with BUSCO[16]. Of the core conserved plant genes, 93.9% and 94.6% were complete in the *I. trifida* and the *I. triloba* genome assemblies, respectively, while another 1.9% and 1.6%, respectively, were fragmented. Collectively, these results demonstrate that our genome assemblies are of high quality.

**Repeat sequences characterization and genome annotation.** Repeat masking with species-specific custom repeat libraries revealed 217.4 Mb (50.2%) of the nongap sequence of the *I. trifida* assembly and 231.1 Mb (52.8%) of the *I. triloba* assembly was repetitive (Supplementary Data 2). For *I. trifida*, the high-confidence gene set consists of 32,301 protein-coding genes represented by 44,158 gene models with an average transcript length of 1662.9 bp and average coding sequence (CDS) length of 1292.9 bp, of which 39,127 (88.6%) were assigned putative functions. For *I. triloba*, the high-confidence gene set consists of 31,423 protein-coding genes represented by 47,088 gene models with an average transcript length of 1784.5 bp and an average CDS length of 1322.7 bp, of which 41,639 (88.4%) were assigned putative functions.

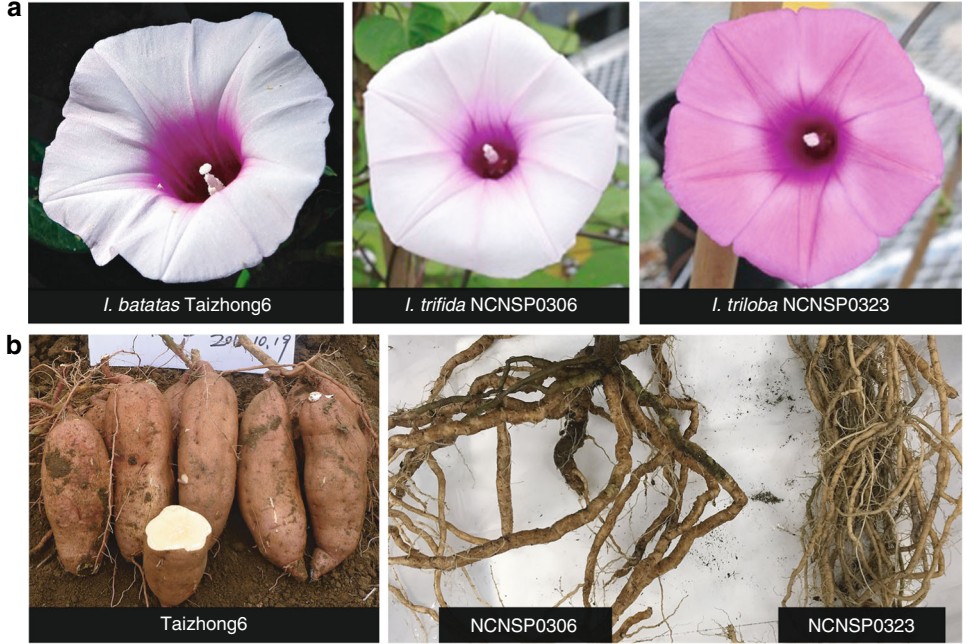

**Fig. 1** Morphology of *Ipomoea batatas*, *I. trifida*, and *I. triloba*. **a**, **b** Flowers (**a**) and roots (**b**) of *I. batatas* "Taizhong6", *I. trifida* NCNSP0306, and *I. triloba* NCNSP0323

**Comparative genomics and lineage-specific family expansion.** We performed comparative analyses using 43,296 protein sequences derived from a transcriptome assembly of *I. batatas* "Beauregard" (Supplementary Method 4) and the predicted proteomes from *I. trifida*, *I. triloba*, and seven other representative plant species, including *I. nil* (Japanese morning glory), *Solanum lycopersicum* (tomato), *Solanum tuberosum* (potato), *Vitis vinifera* (grapevine), *Arabidopsis thaliana* (model eudicot), *Oryza sativa* (rice), and *Amborella trichopoda* (basal angiosperm). A total of 274,278 proteins from the 10 species (79.3% of the input sequences) clustered into 19,901 orthologous groups, of which, 3269 were present only in the four *Ipomoea* species, and 1680 were specific to the Batatas complex (*I. batatas*, *I. trifida*, and *I. triloba*) and absent in *I. nil* (Fig. 2a).

From orthologous gene clustering with other angiosperm proteomes, 73 gene families (orthologous groups) were significantly expanded in *I. trifida* and/or *I. triloba* (Supplementary Data 3). This lineage-specific expansion gene set was enriched with biological processes including "response to stress", "cell death", "response to external stimulus", and "response to biotic stimulus" ($P < 0.0001$, Fisher's exact test; Supplementary Data 4). It is worth noting that two orthologous groups containing genes encoding sporamin proteins were expanded in both *I. trifida* and *I. triloba*. Sporamin, a Kunitz-type trypsin inhibitor (KTI), is a major storage protein in sweetpotato storage roots, and is analogous to patatin in potato which plays an important role in storage, defense, and development[17,18]. Based on the presence of the Kunitz soybean trypsin inhibitor domain (PF00197), we identified 94 and 83 putative KTIs in *I. trifida* and *I. triloba*, respectively, whereas 41, 36, and 16 were present in *I. nil*, potato, and tomato, respectively, and only seven, five, one, and two were found in *Arabidopsis*, grape, rice, and *Amborella*, respectively. Phylogenetic relationships of KTIs in these nine species revealed a subgroup of KTIs specific to *I. trifida* and *I. triloba*, as well as groups of expanded KTIs in potato (Supplementary Fig. 3); suggestive of lineage-specific adaptation events in these root and tuber crops. In particular, two *I. batatas* sporamin genes, orthologous to the *I. trifida* genes *itf10g10920* and *itf01g01870*

that are within the *Ipomoea*-lineage-specific expansion, were upregulated during storage root development in "Beauregard", with expression levels 13–74-fold higher in storage roots compared to fibrous roots (Supplementary Data 5).

**Ipomoea whole-genome triplication.** We identified syntenic blocks within the *I. trifida* and *I. triloba* genomes and inferred whole-genome multiplication events based on the distributions of pairwise synonymous substitution rates ($Ks$) of paralogous genes. The $Ks$ distributions in both *I. trifida* and *I. triloba* showed a peak at 0.65 (Fig. 2b). A substantial proportion of the extant *Ipomoea* genomes occurred in triplicated blocks (23.5%, 34.8%, and 28.0% in *I. trifida*, *I. triloba*, and *I. nil*, respectively), and these proportions are higher than that observed in the tomato genome (6.0%) (Fig. 2c), which bears a signature of whole-genome triplication (WGT)[19]. We further compared the genomes of *I. trifida*, *I. triloba,* and *I. nil*, respectively, with the grape genome. A total of 72.9% of the *I. trifida* gene models are in syntenic blocks corresponding to one grape region. These grape regions collectively cover 79.9% of the grape gene space, among which 36.1% have three orthologous regions in *I. trifida*, 27.1% have two, and 16.8% have one. Similar results were obtained for *I. triloba* and *I. nil* (Supplementary Table 4). Based on the syntenic blocks involving single grape genome segments, each of the *I. trifida*, *I. triloba*, and *I. nil* genomes can be partitioned into three non-overlapping "subgenomes" (Supplementary Fig. 4). Our data indicate that WGT occurred in an ancient ancestor of the *Ipomoea* lineage. While both the tomato and *Ipomoea* WGT events were indicated by peaks of $Ks$ near 0.65, the $Ks$ distribution for tomato, and *I. trifida* orthologous gene pairs is shifted to the right of the paralog-pair distributions for either species (Fig. 2b), indicating that independent WGTs occurred in the ancestry of *Ipomoea* and Solanaceae after their divergence. Based on the evolutionary rate of $7.0 \times 10^{-9}$ substitutions/site/year reported in both Solanaceae[20] and *Arabidopsis*[21], the *Ipomoea* WGT event was estimated to occur around 46.1 million years ago (Mya), much earlier than divergence of *I. nil* from the lineage containing

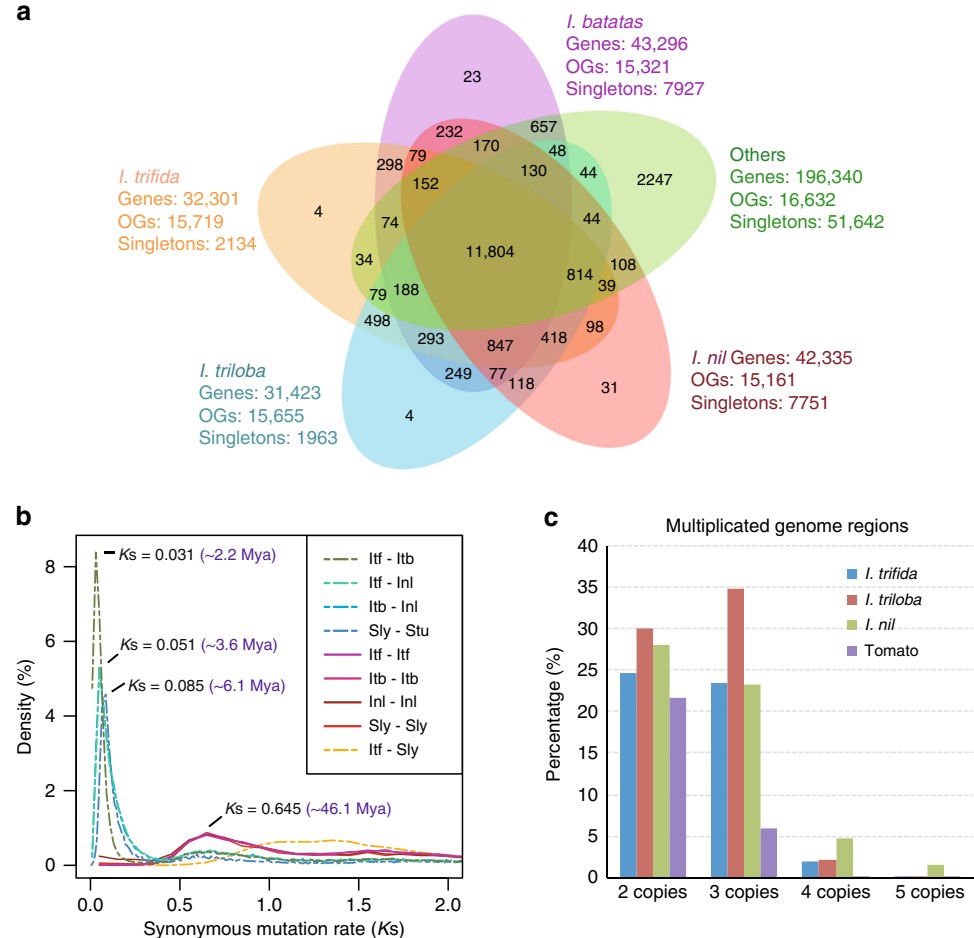

**Fig. 2** Gene family clustering and *Ipomoea* whole-genome triplication. **a** Venn diagram of orthologous gene families. OGs, orthologous groups. 'Others' include tomato, potato, grapevine, *Arabidopsis thaliana*, rice and *Amborella trichopoda*. **b** Distribution of Ks of orthologous or paralogous genes within and between genomes of *I. trifida* (Itf), *I. triloba* (Itb), *I. nil* (Inl), tomato (Sly) and potato (Stu). Estimated times of speciation and WGT events were calculated using a mutation rate of $7 \times 10^{-9}$ substitutions per site per year. Mya, million years ago. **c** Multiplicated genomic regions in *I. trifida*, *I. triloba*, *I. nil*, and tomato

*I. trifida* and *I. triloba* (~3.6 Mya), and the *I. trifida-I. triloba* divergence (~2.2 Mya). Comparison between the genomes of *I. trifida* (or *I. triloba*) and *I. nil* revealed one-to-one syntenic relationships for all 15 chromosomes (Supplementary Fig. 5), suggesting limited large-scale interchromosomal rearrangements over the last 3.6 million years.

A robust gene expression atlas of developmental and stress/hormone treatments (Supplementary Method 4) was used to investigate the fate of gene copies following the ancient WGT event. Genes retained in triplicate or duplicate in the *I. trifida* and *I. triloba* genomes were significantly enriched (*P* < 0.0001, Fisher's exact test) for transcripts that exhibited up-regulation in response to abiotic and biotic stressors, suggesting WGT may play a role in adaptive stress responses. WGT genes were also enriched with gene ontologies suggestive of regulatory functions including "DNA binding", "transcription factor activity", "protein binding", "kinase activity", and "translation regulator activity" (*P* < 0.0001, Fisher's exact test; Supplementary Data 4). For example, two sweetpotato knotted-like homeobox genes, *IBKN2* (orthologous to *itf01g32840*) and *IBKN3* (*itf01g32840*) known to be associated with storage root development[22], were duplicated in the ancestral *Ipomoea* WGT event. In 'Beauregard', these two genes were expressed at higher levels in storage roots as compared with fibrous roots and were upregulated during the development of storage roots

(Supplementary Fig. 6), suggesting that the WGT contributed to additional gene copies that function in storage root development.

**Utility of diploid genomes as references for sweetpotato.** Yang et al.[15] recently presented an 836-Mb haplotype-resolved genome assembly of *I. batatas* "Taizhong6". We assessed the quality of the "Taizhong6" assembly using multiple approaches and found the assembly was incomplete and contained numerous mis-assemblies, limiting its use as a reference for hexaploid sweetpotato (Supplementary Method 5 and Supplementary Figs. 7–12). To evaluate the utility of our two *Ipomoea* diploid genomes as references for hexaploid sweetpotato, we generated ~60× whole-genome sequence data for the hexaploid African landrace "Tanzania" using the 10× Genomics Chromium system (Supplementary Method 6) and aligned the linked reads to the *I. trifida* and *I. triloba* genome assemblies separately; ~83.5% of the reads could be aligned to both assemblies while ~7.7% of the reads did not align (Fig. 3a). Approximately, 5.4% and 3.4% of the reads aligned solely to the *I. trifida* and *I. triloba* genome assemblies, respectively (Fig. 3a). Based on the alignment scores, ~57.7% of the reads aligned better to the *I. trifida* genome than the *I. triloba* genome, while ~31.9% of the reads aligned better in the reciprocal direction (Fig. 3b).

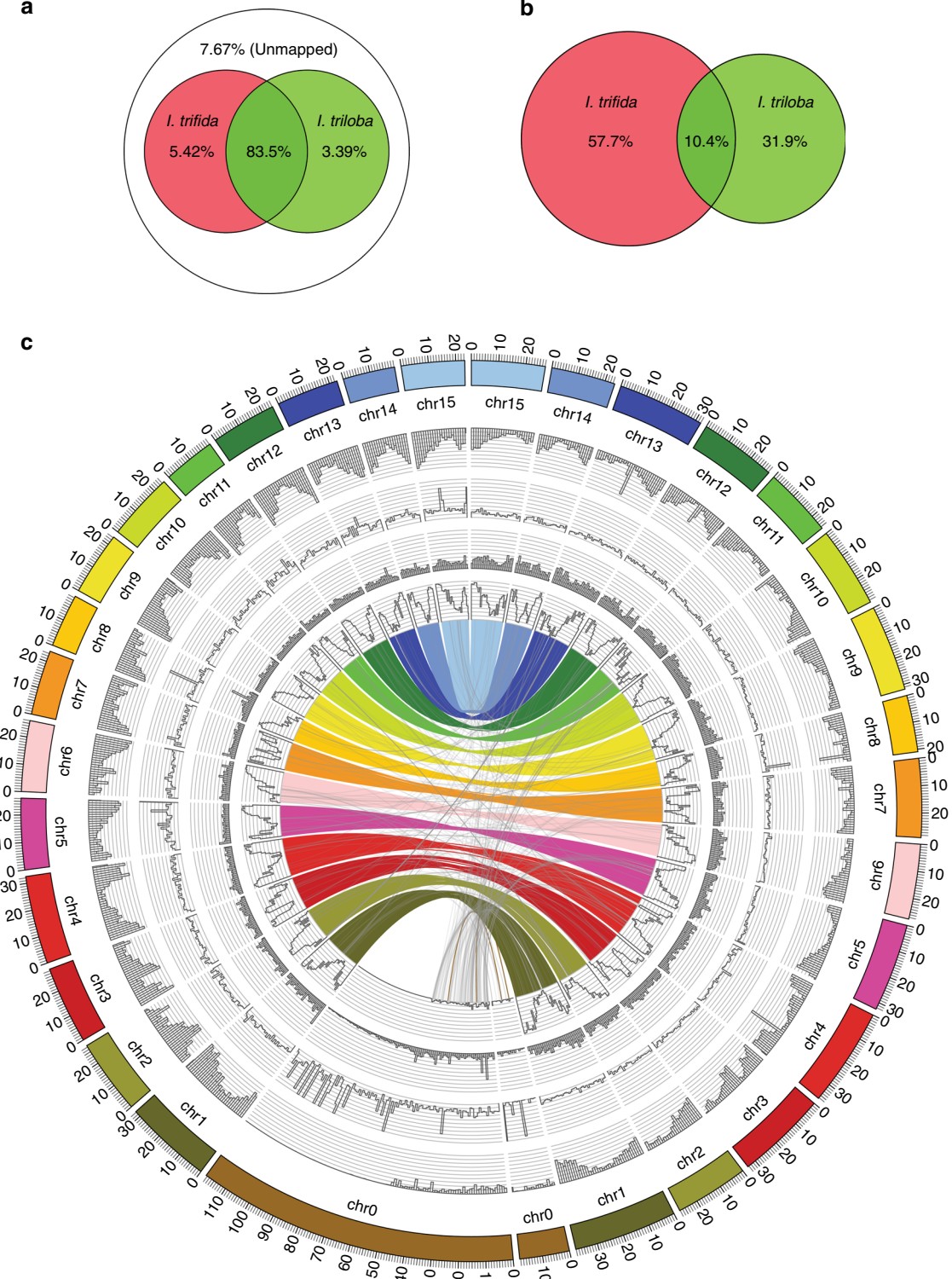

**Fig. 3** Comparative genomic analysis of hexaploid sweetpotato and two wild relatives. **a** Percentages of the mapped and unmapped 10× Genomics linked reads of hexaploid sweetpotato cultivar "Tanzania" to the *I. trifida* and *I. triloba* genome assemblies. **b** Percentages of 10× Genomics reads with better alignments when mapped to one genome assembly compared to the other. **c** Comparison of the hexaploid sweetpotato molecules with the two diploid assemblies. The outermost circle displays ideograms of the pseudochromosomes of the genome assemblies (in Mb scales). The *I. trifida* genome is on the left and the *I. triloba* genome is on the right. The second circle displays the normalized depth of coverage by 10× Genomics reads (1 Mb window). The third circle displays the average read depth of coverage of regions specifically homologous to *I. trifida* or *I. triloba* genomes (1 Mb window). The fourth circle displays the total length of specific regions (1 Mb window size). The fifth circle displays the percentage of the homologous sequences shared among the hexaploid genome and the two diploid genomes (1 Mb window size). The innermost circle displays homologies among the hexaploid and two diploid genomes

Molecules were reconstructed from the linked reads of "Tanzania" using their alignments to the two diploid genome assemblies. Approximately, 4.3 and 4.8 million "Tanzania" molecules of at least 10 kb were identified from alignments to the *I. trifida* and *I. triloba* genome assemblies, respectively, covering ~409 Mb (83%) and ~396 Mb (86%) of the *I. trifida* and *I. triloba* genomes. We identified 390,303 regions in "Tanzania" that were homologous to both diploid genomes, covering ~215 Mb (44%) and ~231 Mb (50%) of the *I. trifida* and *I. triloba* genome assemblies, respectively (Fig. 3c). We further identified regions that were specific to *I. trifida* (~7.0 Mb) or *I. triloba* (~9.9 Mb) genomes, suggesting that the hexaploid sweetpotato genome contains sequences that are uniquely shared with either of these two diploid species (Fig. 3c). Collectively, our results suggest that the two diploid genome assemblies can serve as robust reference sequences for hexaploid sweetpotato as suggested by the high percentages (>90%) of hexaploid reads aligned to either *I. trifida* or *I. triloba*. The *I. trifida* genome is a slightly better reference, while the *I. triloba* genome shares unique features with the hexaploid genome and can serve as a complement to the *I. trifida* genome.

We further analyzed the frequency of the *I. trifida*-type and *I. triloba*-type alleles in "Tanzania". A total of 7.6 million SNP and indel variants were identified between the *I. trifida* and the *I. triloba* genome assemblies, of which, approximately 5.6 million variant sites had at least 10× depth of coverage from the 10× Genomics data of "Tanzania". The majority of these sites (~73.9%) were nonpolymorphic in the "Tanzania" genome comprising both *I. trifida*-specific (~46.2%) and *I. triloba*-specific (~27.7%) homozygous sites (Supplementary Fig. 13a). This observation indicated that both *I. trifida*-like and *I. triloba*-like progenitors had contributed to the "Tanzania" genome, and the *I. trifida*-like progenitor had contributed approximately twice as much as the *I. triloba*-like progenitor. The similar allele depth distribution for the *I. trifida*-specific and *I. triloba*-specific homozygous sites (Supplementary Fig. 13b) in the hexaploid "Tanzania" genome suggested that it was genetic drift that mainly caused the fixation of one allele in all six copies, rather than absence of one genotype in one of the subgenomes leading to four copies of the *I. trifida*-like allele and two copies of the *I. triloba*-like allele. Interestingly, we seldom found "Tanzania" molecules specific to one progenitor type (<1%; i.e., those with all reads supporting only the *I. trifida*-specific or *I. triloba*-specific alleles). Instead, the reads supporting the alleles from different homeologous subgenomes were interwoven within 10× molecules, suggesting recombination between the subgenomes. The proportion of *I. trifida* specific alleles within molecules peaked around 0.6 (Supplementary Fig. 13c), suggesting the approximate amount of contribution of the *I. trifida*-like progenitor genome to the hexaploid sweetpotato genome.

**Facilitating genome-enabled breeding for subsistence farmers.** The Mwanga diversity panel (MDP) contains 16 sweetpotato cultivars, breeding lines and landraces selected from germplasm sourced from different areas across Uganda and a few selected introductions. The MDP accessions have been extensively used as parents in African sweetpotato breeding programs[23]. Previous work using 31 simple sequence repeat (SSR) markers separated the 16 accessions into two genetic groups, population Uganda A and population Uganda B[23]. To revisit their phylogeny and genetic diversity, and to reveal genes and alleles associated with agronomic traits, we resequenced these 16 accessions and aligned the reads to the *I. trifida* and *I. triloba* reference assemblies (Supplementary Data 6). Due to the highly heterozygous nature of the subgenomes within hexaploid sweetpotato, especially

within intergenic regions, accurate variant calling was restricted to genic regions, and a total of 6,138,575 and 6,722,808 high-confidence variant loci were identified in the *I. trifida* and *I. triloba* reference assemblies, respectively, from these 16 accessions (Supplementary Table 5). For both reference genomes, approximately 95% of these loci were bi-allelic. Genotypes of the MDP accessions were consistent with a highly heterozygous genome in hexaploid *I. batatas*, with heterozygous calls at >50% of SNP loci in each accession. The proportions of allele dosage were roughly equivalent among the MDP accessions using either reference genome, at roughly a 3:2:1 ratio for simplex to duplex to triplex dosage classes (Supplementary Fig. 14a, b).

SNP-based genetic clustering of the MDP accessions was similar, but inconsistent with their population membership determined previously using SSR markers[23] (Fig. 4a and Supplementary Fig. 14c), and our results, derived using over one million bi-allelic SNPs with no missing calls for all MDP accessions, represent an improved delineation of their genetic relationships due to the higher resolution available with this high-density, robust polymorphic marker set. Accessions from population A formed two separate and highly supported clusters, whereas population B accessions did not cluster well, with "Magabali", "NASPOT 11", and "New Kawogo" forming a well-supported clade. Principal component analysis (PCA) of the 16 MDP accessions further supported their relationships (Supplementary Fig. 15a), and population structure analysis revealed admixture in most of the 16 accessions (Supplementary Fig. 15b), which was not surprising as several of the lines used in the MDP crossing block can trace their parentage to landraces obtained from farmers in Uganda.

Median read depth was relatively stable from chromosome to chromosome for most accessions, but increased read depth on chromosome 1 initially suggested 6× +1 aneuploidy in "Huarmeyano" and "Mugande", while decreased read depth on chromosomes 8 and 11 suggested 6× −2 aneuploidy in "NASPOT 5" (Fig. 4b and Supplementary Fig. 16). There was no substantial decrease or increase in SNP density for the chromosomes with putative 6× +1 or 6× −2 changes, respectively, suggesting that these chromosomal aberrations in read depth are not likely due to differences in similarity to the reference genomes. Thus, we implemented a statistical test to detect aneuploidy based on read depth (Supplementary Method 7). The test indicated that "NASPOT 5" had a ~0.9 loss of chromosome 8 ($p < 1.5e{-}36$ for ≥5.13 copies) as well as a ~0.6 loss of chromosome 11 ($p < 4.7e{-}78$ for ≥5.39 copies), and "Huarmeyano" had a ~0.3 gain on chromosome 1 ($p < 1e{-}20$ for ≤6.3 copies) and "Mugande" a ~0.5 gain on chromosome 1 ($p < 3e{-}97$ for ≤6.52 copies) (Supplementary Fig. 17). Consistent with our read mapping results, counts of chromosomes in root tip cells confirmed that "NASPOT 5" is a double monosomic line with 88 chromosomes, whereas "Mugande" contains 90 chromosomes (as do "Beauregard" and "Tanzania"), and thus, is not an aneuploid (Fig. 4c and Supplementary Fig. 18). Structural variation in the form of copy number and presence/absence variation have been reported previously in a wide range of crop species and have been associated with phenotypic variation[24–26]. Analyses in potato, another vegetatively propagated polyploid crop, revealed extensive structural variation[27–30], including presence/absence variation of sequences up to 575 kb in length that impacts transcript dosage. The discovery of aneuploidy in cultivated sweetpotato may present an extreme form of structural variation that clearly would affect transcript dosage and consequently, phenotypic variation.

Orange flesh in sweetpotato is correlated with high β-carotene content, and cultivation and consumption of OFSP has been promoted in SSA to help alleviate vitamin A deficiency[31].

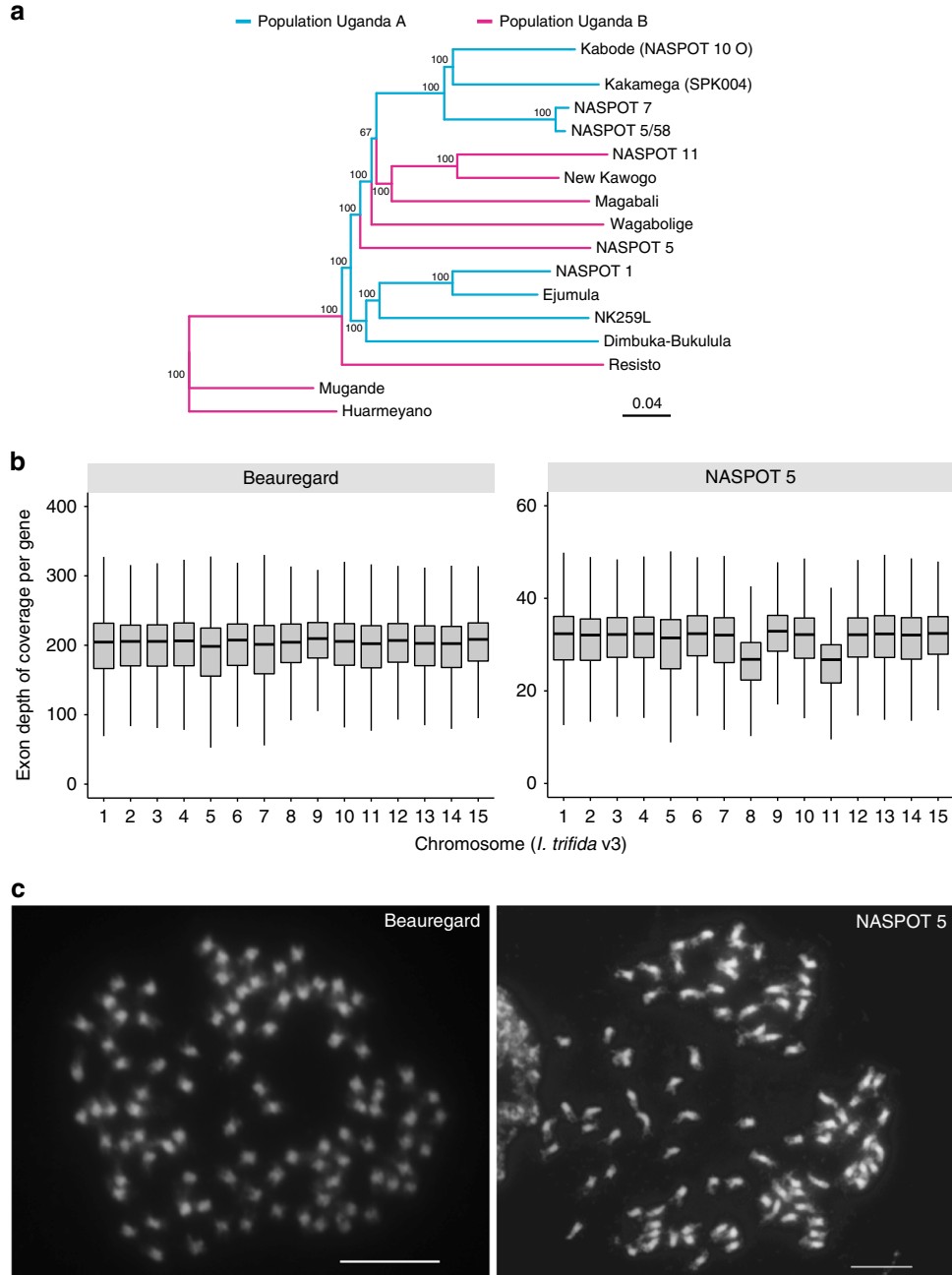

**Fig. 4** Genetic diversity of key African sweetpotato accessions. **a** Neighbor-joining phylogenetic tree of the Mwanga Diversity Panel accessions based on single nucleotide polymorphisms detected using the *l. trifida* genome as the reference. Population membership based on previous simple sequence repeat analysis is indicated by branch colors. Numbers at nodes indicate the percentage of 1000 bootstrap replications that support each clade. Huarmeyano was used as an outgroup for rooting. **b** Read depth analyses of "Beauregard" and "NASPOT 5" based on alignment to the *l. trifida* genome. For each box plot, the lower and upper bounds of the box indicate the first and third quartiles, respectively, and the center line indicates the median. **c** A metaphase cell of "Beauregard" showing 90 chromosomes and "NASPOT 5" showing 88 chromosomes. Bars represent 10 μm

However, other traits such as yield, dry matter content, disease and insect resistance, and stress tolerance are important in development of consumer-accepted varieties. The MDP is composed of orange- and white-fleshed accessions (Supplementary Data 6), and to enable stacking high β-carotene with other agronomic traits, SNPs associated with flesh color in the MDP were identified. The identified SNPs occurred in homologs of genes encoding key carotenoid biosynthetic enzymes, including phytoene synthase (PSY), phytoene desaturase (PDS), ζ-carotene isomerase (Z-ISO), and lycopene β-cyclase (LCYB) (Fig. 5a, b, Supplementary Data 7 and Supplementary Method 8). The *PSY*

(*itf03g05110*), *PDS* (*itf11g08190*) and *Z-ISO* (*itf04g12320*) genes were of particular interest as they were upregulated in orange-fleshed 'Beauregard' storage roots as compared to other root types (Fig. 5c). In contrast, the *LCYB* (*itf04g32080*) gene had reduced expression in storage roots as compared to fibrous roots, suggesting it may not be involved in conferring orange flesh in "Beauregard" storage roots. Furthermore, three SNPs in *PSY* that were significantly associated with orange flesh in the MDP showed biased expression of the "orange" alleles in "Beauregard" storage roots at 40 and/or 50 days after transplanting (DAT) (Supplementary Fig. 19), suggesting that the basis of increased

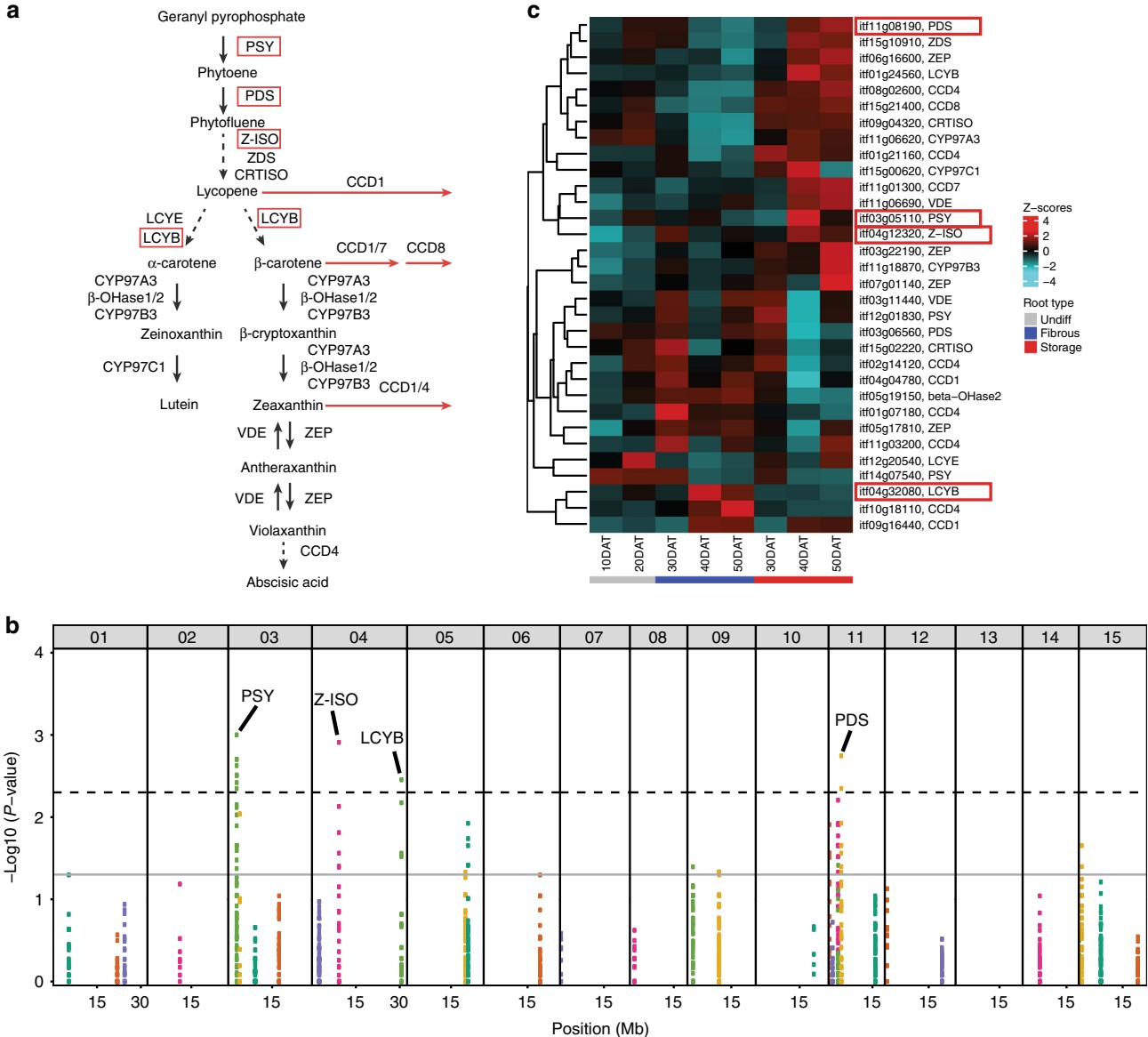

**Fig. 5** Analysis of carotenoid metabolism in hexaploid sweetpotato. **a** Carotenoid biosynthesis and degradation pathways. Dashed arrows indicate multi-step reactions. Red arrows indicate degradation reactions. Red rectangles indicate components with significantly enriched SNPs at $p < 0.005$ (Fisher's exact test). PSY, phytoene synthase; PDS phytoene desaturase; Z-ISO ζ-carotene isomerase; ZDS ζ-carotene desaturase; CRTISO carotenoid isomerase; LCYE lycopene ε-cyclase; LCYB lycopene β-cyclase; β-OHase β-ring hydroxylase; CYP cytochrome P450; VDE violaxanthin de-epoxidase; ZEP zeaxanthin epoxidase; CCD carotenoid cleavage dioxygenase. **b** Fisher's exact test for differences in allele frequencies between orange and white-fleshed MDP accessions for putative carotenoid biosynthesis loci. Each color within a chromosome indicates SNPs from the same gene. The dotted and solid lines indicate $p = 0.005$ and $p = 0.05$, respectively. **c** Expression profiles of genes involved in carotenoid biosynthesis in different types of roots of 'Beauregard' during development. Gene-wise Z-scores were calculated from the arithmetic means of replicates after log2 transformation of FPKM values plus 1. DAT days after transplanting; Undiff undifferentiated

carotenoid accumulation in "Beauregard" is expression-based. One SNP associated with orange flesh in *PDS* was weakly significant for biased expression of the "orange" allele at 20 DAT yet not significant in 50 DAT storage roots, suggesting that expression of specific *PDS* alleles prior to storage root initiation might be important in the development of orange flesh. Together, our analysis identified logical control points for the accumulation of β-carotene in OFSP, and these loci provide targets for marker-assisted selection of orange-fleshed varieties.

**Origins and evolution of the Batatas complex**. Phylogenetic inferences attempting to describe the evolutionary history of the Batatas complex have been highly incongruent[4,6,32,33], yet no

study to date has explicitly tested for hybridization or reticulate evolution within the complex. Rather than a strictly bifurcating tree, the true evolutionary relationships among species or populations within the Batatas complex may be best understood as a network with reticulations[34,35]. To test the hypothesis of recent or ancient hybridization in the Batatas complex, we performed targeted sequence capture on 490 genes that are single copy in the *I. trifida* and *I. triloba* genomes as well as genome assemblies of other core eudicots as described in *Amborella* Genome Project[36]. We first included twenty-six accessions in a species-tree analysis (Supplementary Method 9 and Supplementary Data 8), representing the breadth of phenotypic diversity across the Batatas complex. Our sample included *I. triloba* and *I. trifida* accessions

with genome sequences (NCNSP0323 and NCNSP0306; ITR and ITRk[14]), and twelve polyploid *I. batatas* accessions. Haplotype diversity was assessed for each gene within each of the *I. batatas* assemblies and only genes with a single haplotype (i.e., nulliplex) or two haplotypes that were sister in the gene trees were included in phylogenetic analyses. The coalescence-based species-tree estimation did not account for reticulations in the phylogeny, but nonetheless recovered six distinct lineages within the Batatas complex (Fig. 6a and Supplementary Fig. 20). *Ipomoea triloba* accessions formed a clade, whereas relationships could not be resolved within a clade including all *I. trifida* and cultivated sweetpotato (*I. batatas*) accessions, and an unnamed diploid accession. Another unnamed diploid accession was recovered as sister to all other members of the Batatas complex aside from *I. splendor-sylvae* (Fig. 6a). While there was no overlap in accessions used in this study and a recently published analysis of Batatas complex species relationships based on 307 exon-capture assemblies and the whole plastid genome[6], both analyses infer early divergence of ancestral lineages for *I. splendor-sylvae* and the remainder of the Batatas complex, followed by divergence of lineages leading the last common ancestors of accessions in the *I. batatas* + *I. trifida* clade and the *I. triloba* + *I. lacunosa* + *I. cordatotriloba* + *I. tenuissima* clade.

Reticulations in the phylogeny were tested using pseudolikelihood method for inferring phylogenetic networks[37]. The PhyloNet[37] analysis was run using a single exemplar for polyploid *I. batatas* together with all of the diploid Batatas complex accessions used in the species-tree analysis. The likelihood of a network with two reticulations (Fig. 6b) was significantly better than a tree with the same samples estimated without reticulations ($p < 0.001$, likelihood ratio test; Supplementary Fig. 20d), but otherwise the relationships among cultivated sweetpotato (using 'Tinian' as an exemplar), *I. trifida* and *I. triloba* remained unchanged. The reticulation giving rise to cultivated sweetpotato exhibited evidence of a wide ancestral cross. However, since the taxonomy of accession PI 552786 collected from Bolivia is uncertain, the results do not clearly favor conspecific parents (autopolyploidy) or interspecific hybridization (allopolyploidy) within the *I. trifida* + *I. batatas* clade as giving rise to polyploid *I. batatas*. The deeper reticulation in the PhyloNet network suggests that extensive gene flow between divergent ancestral populations gave rise to the clade including *I. triloba*, *I. lacunosa*, *I. cordatotriloba* and *I. tenuissima*. With respect to *I. batatas* "Tinian", the inference that parental lineages were genetically divergent (Fig. 6b) was consistent with results of mapping the *I. batatas* "Tanzania" genomic reads to the *I. trifida* and *I. triloba* reference genomes and allele frequency analysis (Fig. 3 and Supplementary Fig. 20). This finding underscores the continuum between autopolyploidy and allopolyploidy in outcrossing species.

## Discussion

Sweetpotato is an important food security crop with a highly recognized potential to alleviate hunger, vitamin A deficiency, and poverty in SSA. We sequenced the genomes of two diploid relatives of sweetpotato, *I. trifida* and *I. triloba*, and demonstrated that these high-quality assemblies can be used as robust references to facilitate sweetpotato breeding. While *I. trifida* is the more closely related diploid to hexaploid sweetpotato, and is a better reference for sweetpotato, the *I. triloba* genome can be a complementary reference sequence.

Many species in the Batatas complex are able to hybridize with one another, and hybridization seems to be important in the evolution of Batatas complex populations[38–40]. Some diploid species within the complex have been hypothesized as hybrid species, and sweetpotato is thought to have arisen through either

allopolyploidization[4], autopolyploidization[6], or both[10]. Furthermore, microsatellite data provide evidence supporting on-going gene flow between sympatric populations of *I. cordatotriloba* and *I. lacunosa*[41]. The observation that many species in the Batatas complex appear morphologically similar and show evidence of hybridization may be the result of ongoing or ancient hybridization. Results from read mapping and phylogenetic analyses suggest that cultivated sweetpotato is derived from a cross between divergent parents but their taxonomy is equivocal.

Comparative analyses using the diploid genome sequences revealed an ancient WGT event specific to the *Ipomoea* genus. We identified gene families specifically expanded in *I. trifida* and/or *I. triloba* associated with adaptation and stress resistance in roots including sporamin genes, which could be a prerequisite for storage root development in the hexaploid sweetpotato. Genome resequencing of 16 sweetpotato cultivars and landraces used in African breeding programs allowed refinement of the genetic relationships among these key accessions, an understanding of dosage distribution within cultivated sweetpotato, and identification of genes and alleles associated with high β-carotene content, highlighting how genomic tools can enable more efficient improvement of sweetpotato. We anticipate that these resources will become increasingly valuable for the global sweetpotato breeding and genomic community as we seek to further improve this critical food crop using genome-enabled marker-assisted breeding methods.

## Methods

**De novo genome assembly.** For *I. triloba* NCNSP0323, high-quality cleaned Illumina paired-end and mate-pair reads (Supplementary Method 1) were assembled into scaffolds with SOAPdenovo2 (ref. [42]) and gaps in the resulting scaffolds were filled with GapCloser in the SOAPdenovo2 package. For *I. trifida* NCNSP0306, cleaned paired-end and mate-pair reads were assembled into scaffolds using Platanus[43] (v1.2.1). The error-corrected PacBio long reads were used to further fill gaps in the scaffolds and to connect scaffolds using PBJelly[44] (v15.8.24). Pilon[45] (v1.13) was used to improve the assembly by correcting base errors, fixing misassemblies, and further filling gaps. Potential contamination from microorganisms was detected by aligning the assemblies to the NCBI nonredundant nucleotide (nt) database using BLASTN with an *e* value cutoff of 1e−5. Scaffolds with more than 90% of their length similar to only bacterial or virus sequences were considered contaminants and removed. Finally, scaffolds contained within other scaffolds with sequence identity >99% and coverage >99% were removed.

**Genome annotation.** Custom repeat libraries were individually created for *I. trifida* and *I. triloba* by combining the putative repeat libraries predicted from MITE-Hunter[46] (v2011) and RepeatModeler (http://www.repeatmasker.org/; v1.0.8). Protein-coding genes were removed from the repeat library using ProtExcluder[47]. The Repbase (v20150807) repeats for green plants (Viridiplantae) were then added to each library to create a final custom repeat library for each species. The pseudomolecules for *I. trifida* and *I. triloba* were repeat masked with the respective repeat library using RepeatMasker (http://www.repeatmasker.org/; v4.0.6).

For gene prediction, AUGUSTUS[48] (v3.1) was trained on the soft-masked assemblies using the leaf RNA-Seq alignments. Gene models were predicted with AUGUSTUS using the hard-masked assemblies and refined with PASA2 (v2.0.2) (ref. [49]) using the genome-guided transcript assemblies from all tissue as transcript evidence (Supplementary Method 1). Two rounds of gene prediction comparison were performed and gene models PASA identified as merged, but unable to split, were manually inspected and split as necessary. A third round of gene prediction comparison was performed to refine the structure of the manually curated models. The final output from PASA2 is the working set of gene models. Expression abundances for each gene model were determined based on the RNA-Seq read alignments using Cufflinks2 (ref. [50]) (v2.2.1). A high-confidence gene model set was constructed from the working gene model set by removing partial gene models and gene models with an internal stop codon, a hit to a transposable element, or an FPKM of 0 across the RNA-Seq libraries used for the annotation.

Functional annotations for the high-confidence gene models were assigned by comparing their protein sequences against the Arabidopsis proteome (TAIR 10), Pfam (v29), and the Swiss-Prot databases. Proteins that only matched a hypothetical Arabidopsis gene model and had no matches in the other databases were annotated as conserved hypothetical, while proteins with no matches in any of the databases were annotated as hypothetical.

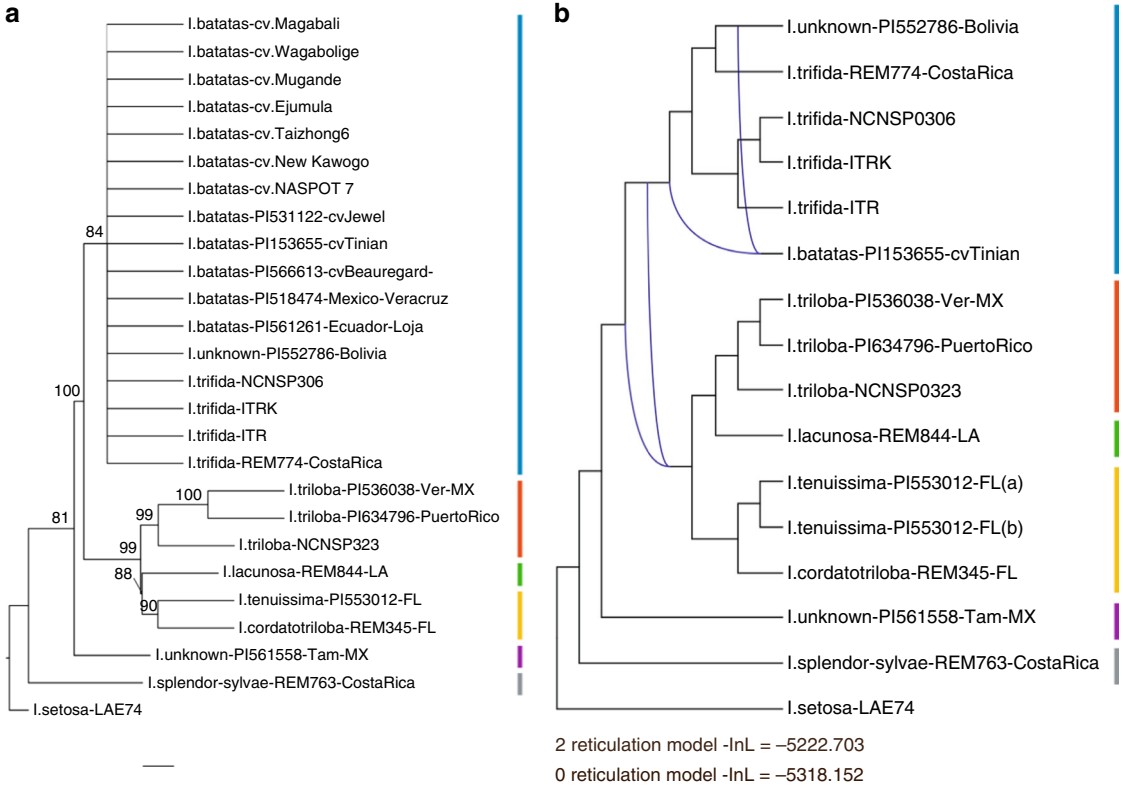

**Fig. 6** Phylogenetic relationships among members of the Batatas complex. **a** Phylogenetic relationships among Batatas complex accessions inferred using the species-tree method ASTRAL-II. Bootstrap support values are shown next to the branches. Relationships among *I. trifida* and *I. batatas* accessions were not resolved. **b** Phylogenetic network of the Batatas complex including pseudolikelihood estimates for hybridization events in the ancestor of red, green and yellow clades as well as in the ancestry of cultivated sweetpotato. Blue lines indicate the two-reticulation events inferred in PhyloNet. Below the phylogenetic network are −log-likelihood scores for the tree generated in PhyloNet under the zero-reticulation model (Supplementary Fig. 20d) and the network estimated under the two-reticulation model. Color bars at the right of each tree denote the six inferred major lineages

**Comparative genomics analysis and gene family expansions.** OrthoFinder[51] was used to infer homology among proteins from *I. batatas* (derived from the "Beauregard" transcriptome assembly; Supplementary Method 4), *I. trifida*, *I. triloba*, and seven additional plant species: *I. nil* (v1.1), potato (*S. tuberosum*; v3.4), tomato (*S. lycopersicum*; ITAG2.4), grape (*V. vinifera*; Genoscope 12×), Arabidopsis (*A. thaliana*; TAIR10), rice (*O. sativa*; v7.0), and Amborella (*A. trichopoda*; v1.0).

Lineage-specific gene family expansion was inferred from orthologous groups using the software CAFE[52] (v3.0). *I. batatas* was excluded from the analysis because its transcriptome assembly did not represent the complete gene content. The random gene birth and death rates were estimated using a maximum likelihood method[52] across the species tree composed of the nine species. Orthologous groups with accelerated rate of expansion in *I. trifida* and/or *I. triloba* were determined with a branch-specific Viterbi *p* value cutoff of 0.01. Given that CAFE may fail to provide accurate lambda estimates for gene families with large differences in size among species, we also considered gene families expanded if they contained more than ten genes in either *I. trifida* or *I. triloba* and at least 50% more genes in these two species than any of the other investigated species. Fisher's exact test followed by a Bonferroni multiple test correction was used to determine gene enrichment within 96 gene ontology terms as well as for tissue specific, biotic and abiotic stress conditions.

**Synteny and whole-genome duplication analysis.** To identify syntenic regions within and between genomes, protein sequences of *I. trifida*, *I. triloba*, *I. nil*, tomato, and potato were self-aligned and aligned with each other using BLASTP. High-confidence collinear blocks were determined using MCScanX[53] with an *e* value cutoff of 1e−10. *Ks* values of homologous pairs were calculated using the Yang–Nielsen algorithm implemented in the PAML package[54]. Divergence time (*T*) was calculated as $T = Ks/2r$, where *r* is the evolutionary rate ($7.0 \times 10^{-9}$ substitutions/site/year), according to those reported in the Solanaceae[20] ($6.96 \times 10^{-9}$) and Arabidopsis[21] ($7.0 \times 10^{-9}$).

**10× Genomics data analysis.** We generated 10× Genomics sequences for hexaploid sweetpotato cultivar "Tanzania" (Supplementary Method 6). To reconstruct

"Tanzania" molecules sequenced by the 10× platform, we grouped the reads by barcodes. For each group, we then identified clusters of linked reads, which were mapped to the same region in the reference genome. Two consecutively mapped reads were considered linked (i.e., sequenced from the same "Tanzania" molecule) if their barcodes were identical and the alignment distance between them was ≤10 kb. Each linked read cluster was regarded as a span of a "Tanzania" molecule. Only clusters spanning at least 10 kb in the reference genome were retained. It should be noted that, given the aforementioned method for the molecule reconstruction, each "Tanzania" molecule represented a homologous region between the "Tanzania" genome and the diploid genome that was used as the reference. If a "Tanzania" molecule is homologous to both diploid genomes, i.e., the two linked read clusters for a "Tanzania" molecule obtained from using the two diploid reference genomes are identical, then we considered it to be a three-way homology meaning the homologous region was shared between the "Tanzania" genome and the two diploid genomes.

**Variant detection and phylogeny of hexaploid sweetpotatoes.** Whole genome sequence reads from the 16 MDP accessions (Supplementary Method 7) were cleaned using Cutadapt[55] (v1.11) with a base quality threshold of 20 and a minimum length of 200 bases after trimming. Reads were aligned to the *I. trifida* and *I. triloba* genome assemblies using BWA-MEM[56] (v0.7.12) in paired-end mode with default parameters. Picard tools (v2.7.1; http://broadinstitute.github.io/picard) were used to remove duplicate read pairs and alignments with a MAPQ score less than 30. Alignments were refined using the GATK IndelRealigner[57] (v3.6.0). Alignments to the same reference assembly from all individuals were analyzed together for variant and genotype calling at regions overlapping genes only. FreeBayes[58] (v1.1.0) was used with an assumed ploidy of 6, "haplotype-length" of 0, minimum base quality of 20, and a minimum of 2 reads and 10% of the reads in at least one single individual supporting an alternate allele. Variant records were filtered using the "vcffilter" function in vcflib (v1.0.0; https://github.com/vcflib/vcflib) with a minimum variant quality score of 40, "SAF" > 1, "SAR" > 1, "RPR" > 1, and "RPL" > 1. Individual genotypes were filtered using a minimum depth of 18, maximum depths based on average depth across genic regions plus 2 standard deviations for each sample, and a minimum of 5% of reads supporting the least frequent allele in

a genotype call. Complex variants were decomposed to SNPs and indels using the "vcfallelicprimitives" function in vcflib and variant records were removed if all the genotypes containing the alternate allele failed the above quality filters. All analyses were performed using these filtered variant and genotype calls, with additional processing as described.

Phylogenetic analysis was performed using bi-allelic SNPs with no missing data for any accessions, with Rphylip[59] (v0.1–23). Nei's genetic distances were calculated using 'gendist' and a neighbor-joining tree was constructed using 'neighbor'. Bootstrap analysis was done using 1000 replications created with 'boot.phylo' from the R package, ape[60] (v4.1). PCA analysis was performed using PLINK[61] (v1.9) and population structure analysis using STRUCTURE[62] (v2.3.4). The most likely number of clusters (K) was estimated by calculating delta K[63]. STRUCTURE analyses were run 20 times for each K value ranging from 1 to 10, using 8,000 randomly selected SNPs with an admixture model. The subgroup memberships of accessions were determined based on 10,000 iterations.

**Phylogenomic analysis of species in the Batatas complex**. We performed targeted sequence capture on 490 single-copy genes for fifteen accessions in the Batatas complex and computationally extracted target exon sequences from available genome assemblies for *I. trifida* (NCNSP0306, ITR and ITRk[14]) and *I. triloba* (NCNSP0323), and from shotgun reads from 18 *I. batatas* accessions (Taizhong6[15], Tanzania, and 16 MDP accessions) (Supplementary Method 9). Multiple sequence alignment for each single-copy gene family was performed using PRANK[64], and alignments were filtered using Gblocks[65] to remove poorly aligned regions. Gene trees were estimated separately for each gene alignment in RAxML with 100 bootstraps[66] (v8.0.0). For each gene, samples exhibiting more than a single-hapotype or an accession-specific clade in the gene tree were removed (Supplementary Method 9). After filtering, 11 hexaploid *I. batatas* accessions had fewer than 30 retained single-hapotype genes and were removed from further analysis. A majority of the remaining 12 *I. batatas* accessions were retained in 85 gene trees. These single-copy/single-haploid gene trees were included in ASTRAL-II[67] (v4.10.12) and SVDQuartets[68] estimates of relationships among the Batatas-complex species and accessions while accounting for incomplete lineage sorting.

PhyloNet[37] was used to infer a phylogenetic network while accounting for incomplete lineage sorting under zero, one, two, and three reticulation scenarios using the InferNetwork_MPL option and gene tree from RAxML as the input. Five independent analyses of each reticulation scenario were carried out to best traverse the complex parameter space. The phylogenetic network with the best likelihood score, which was significantly different from the likelihood of a zero-reticulation network (i.e., the ASTRAL-II and SVDQuartets trees) was chosen as the most likely network given the gene trees. PhyloNet is computationally intensive, so only the 'Tinian' accession was included as an exemplar for *I. batatas*.

## Data availability

The genome assemblies and raw genome reads of *I. trifida* and *I. triloba* have been deposited into GenBank BioProject under accessions PRJNA428214 and PRJNA428241, respectively. Raw 10x Genomics, genome resequencing and RNA-Seq reads are available in the National Center for Biotechnology Information Sequence Read Archive under accessions SRP161954, SRP162006, SRP132113, SRP132112, SRP162112, SRP162110, and SRP162021. The pseudomolecules, genome annotation, expression abundances, BLAST server, and Jbrowse instance are available in the Sweetpotato Genomics Resource (http://sweetpotato.plantbiology.msu.edu). In addition, the genome assemblies, annotated genes, expression abundances, and variants of the MDP are available via the Dryad Digital Repository (https://doi.org/10.5061/dryad.b9m61cg). Results of phylogenetic analysis and the associated multiple sequence alignments were available in the Github repository (https://github.com/laeserman/Wu-et-al.-2018-Nature-Comm).

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

## Acknowledgments

We are grateful to Maria del Rosario Herrera, Federico Diaz, Veronica Mosquera and Maria David for technical support, and to Drs. Mingcheng Luo and Tingting Zhu for help in generating BioNano genome maps. This research was supported by grants from the Bill & Melinda Gates Foundation (OPP1052983), National Natural Science Foundation of China (31461143017), National Key Research and Development Program of China (Minor cereal Fund), National Science Foundation (DEB-1601251), The North Carolina SweetPotato Commission, and the North Carolina State University Agricultural Research Service. Research at CIP was undertaken as part of the CGIAR Research Program on Roots, Tubers and Bananas (RTB) and supported by CGIAR Fund Donors (http://www.cgiar.org/about-us/our-funders/). This research was also supported by the use of the NeCTAR Research Cloud, by QCIF and by the University of Queensland's Research Computing Centre (RCC). The NeCTAR Research Cloud is a collaborative Australian research platform supported by the National Collaborative Research Infrastructure Strategy.

## Author contributions

Z.F., C.R.B., G.C.Y., L.J.M.C, J.L.M, A.K., Q.C. and J.K. designed and managed the project. G.C.Y., R.O.M.M., C.J., D.C.G., A.K., M.K., M.G., G.T.G. and K.W.-R. prepared samples and extracted DNA/RNA for genome and transcriptome sequencing. Q.C., D.M., X.Y. J.S., G.C.Y. and Z.F. contributed to genome and transcriptome sequencing. S.W., H.S., Q.C. and K.B. performed genome assembly and comparative genomic analysis. J.P.H., G.T.G., E.C., B.V., K.W-R. and S.W. performed genome annotation and transcriptome analysis. K.H.L., J.P.H., N.M-C., X.W. and L.J.M.C conducted analysis of genome resequencing data and population genomics analysis. W.G., B.A.O., D.C.G., A.K. and C.Z. developed and sequenced the diploid mapping population and constructed the genetic map. C.Z. and L.J.M.C. generated the 10× Genomics data. L.E. and J.L.M. conducted targeted sequencing and phylogenomic analysis. H.W. and J.J. performed chromosome counting. S.W., K.H.L., Z.F. and C.R.B. wrote the paper. All authors read and approved the manuscript.

## Additional information

**Competing interests:** The authors declare no competing interests.

