## [Peer Review File · Nature Communications]

REVIEWERS' COMMENTS:

Reviewer #1 (Remarks to the Author):

Comments on revision of sweet potato manuscript, Nature Communications.

On the WGT, I haven't checked back to the other paper noted, but I trust the report noted a WGD as opposed to a WGT. On the Ks plot, I think it should be further improved by using a log₁₀ scale for the X axis. Alternatively, show this one for the range Ks = 0-0.5, and then log₁₀ for the rest. Distinction and clarity on the peak at 0.645 is important.

Regarding the phylogenetic analysis, the inclusion of new genotypes is a good thing. However, the resolution of the large species-tree lineage supported at 84 should be better teased out, as this might be very interesting concerning interrelationships between the diploids and hexaploids contained therein.

So, re: "Despite aggressive filtering of loci and samples with retention of divergent haplotypes, extensive gene tree discordance was observed within the *I. trifida* + *I. batatas* clade and we were not able to resolve relationships among accessions within the clade." and "Our phylogenetic analyses were not able to distinguish, however, whether the parental lineages were divergent genotypes within a single species that was the ancestor of the *I. trifida* + *I. batatas* clade (autopolyploidy) or two species.

...I think the authors should present careful quantitative results on different gene tree topologies linking the hexaploids and diploids in the 84 clade. Patterns there, despite not resolvable in the species tree, could be extremely interesting. Perhaps also a DensiTree "cloudogram" would be illustrative.

Point 9 from their response to my review: nicely done.

Point 10 from the response to my review: looks safe to me. I haven't looked; is this Fig. 1 now included in the supplement and the argumentation that accompanies it? I understand that the authors are writing a response to Muñoz-Rodríguez et al. but I think the results are relevant in this paper in order to be meticulous in presentation of "single copy genes" for readers of a high-profile journal.

Point 11 from the response: I'm sorry but I don't buy it. My comment about improved likelihood from adding reticulations seems to have been given short shrift. I do understand that the authors are trying to say there was a plateau in likelihood with addition of more reticulations, but I still see no objective reason to prefer two over 3, other than if the authors wish to add in a parsimony argument... i.e., likelihood plus parsimony in terms of reticulations until plateau. Sort of something like the Akaike information criterion. Isn't this

what you're actually trying to say? Please spell this out better.

Point 13: good.

Point 3 for reviewer #2 - excellent to show 3:1 for Ipomoea relative to Vitis.

Reviewer #2 (Remarks to the Author):

Wu et al. "Genome sequences of Ipomoea trifida and Ipomoea triloba, two diploid wild relatives of cultivated sweetpotato, reveal targets for improvement of a key food security crop".

I was one of the reviewers for a previous draft of this paper so I have directed my focus towards the authors' response to the queries that myself and other reviewers raised. From my end, most previously weak points are addressed in this version – so I am supportive of the publication this time.

Reviewer #1 (Remarks to the Author):

Comments on revision of sweet potato manuscript, Nature Communications.

On the WGT, I haven't checked back to the other paper noted, but I trust the report noted a WGD as opposed to a WGT. On the Ks plot, I think it should be further improved by using a log10 scale for the X axis. Alternatively, show this one for the range Ks = 0-0.5, and then log10 for the rest. Distinction and clarity on the peak at 0.645 is important.

Response: Thanks for the reviewer's suggestion. We changed the break interval in the histogram from 0.02 to 0.1, and the distinction and clarity of the 0.645 peak have been improved. We also tried a log10 transform for the X axis (right), but it is complicated to put untransformed (Ks = 0 to 0.5) and transformed ($\log_{10}Ks = -0.5$ to 0.5) in one plot, so this one was not adopted.

Regarding the phylogenetic analysis, the inclusion of new genotypes is a good thing. However, the resolution of the large species-tree lineage supported at 84 should be better teased out, as this might be very interesting concerning interrelationships between the diploids and hexaploids contained therein.

So, re: "Despite aggressive filtering of loci and samples with retention of divergent haplotypes, extensive gene tree discordance was observed within the *I. trifida* + *I. batatas* clade and we were not able to resolve relationships among accessions within the clade." and "Our phylogenetic analyses were not able to distinguish, however, whether the parental lineages were divergent genotypes within a single species that was the ancestor of the *I. trifida* + *I. batatas* clade (autopolyploidy) or two species.

...I think the authors should present careful quantitative results on different gene tree topologies linking the hexaploids and diploids in the 84 clade. Patterns there, despite not resolvable in the species tree, could be extremely interesting. Perhaps also a DensiTree "cloudogram" would be illustrative.

Response: As expected in the face of short periods between speciation events and hybridization, there is extensive discordance among individual gene trees. ASTRAL used in our study accounts for incomplete fixation of ancestral alleles (i.e. Incomplete Lineage Sorting, "ILS") between speciation events and PhyloNet (also used in our study) accounts for both ILS and hybridization. Unfortunately, in order to display the variation among gene trees using DensiTree as suggested by the reviewer, all samples must be included in each gene tree and that is not the case with our data. Therefore, we have deposited machine readable trees (newick format) and PDF versions of each gene tree in the Github repository with our multiple sequence alignments (<https://github.com/laeserman/Wu-et-al.-2018-Nature-Comm>).

Point 10 from the response to my review: looks safe to me. I haven't looked; is this Fig. 1 now included in the supplement and the argumentation that accompanies it? I understand that the authors are writing a response to Muñoz-Rodríguez et al. but I think the results are relevant in this paper in order to be meticulous in presentation of "single copy genes" for readers of a high-profile journal.

Response: This figure has been added to Supplement (Figure S16a) and cited in the Supplementary Methods (Page 39).

Point 11 from the response: I'm sorry but I don't buy it. My comment about improved likelihood from adding reticulations seems to have been given short shrift. I do understand that the authors are trying to say there was a plateau in likelihood with addition of more reticulations, but I still see no objective reason to prefer two over 3, other than if the authors wish to add in a parsimony argument... i.e., likelihood plus parsimony in terms of reticulations until plateau. Sort of something like the Akaike information criterion. Isn't this what you're actually trying to say? Please spell this out better.

Response: Right, but rather than using Akaike information criterion (AIC) or Bayesian information criterion (BIC), we used a likelihood ratio test to assess whether any observed improvement in the log pseudo-likelihood scores were statistically significant. As shown in Figure 6, the logs of the pseudo-likelihood scores for 0 and 2 reticulations are -5318.15 and -5227.70, respectively ($p \ll 0.01$). The log pseudo-likelihood for three reticulations, -5298.18, was actually worse than for two reticulations. A drop in the pseudo-likelihood with additional reticulations was also observed in yeast gene trees (Yu & Nakhleh 2015; <https://doi.org/10.1186/1471-2164-16-S10-S10>). In both cases, the gene trees did not provide signal for additional reticulations.